# The Prevalence and Severity of External Auditory Exostosis in Young to Quadragenarian-Aged Warm-Water Surfers: A Preliminary Study

**DOI:** 10.3390/sports8020017

**Published:** 2020-02-04

**Authors:** Vini Simas, Wayne Hing, James Furness, Joe Walsh, Mike Climstein

**Affiliations:** 1Water Based Research Unit-Bond Institute of Health and Sport, Bond University, Gold Coast, QLD 4226, Australia; 2Independent Researcher, Concord, NSW 2137, Australia; 3School of Health and Human Sciences, Southern Cross University, Bilinga, QLD 4225, Australia; 4Physical Activity, Lifestyle, Ageing and Wellbeing Faculty Research Group, University of Sydney, Sydney, NSW 2006, Australia

**Keywords:** auditory exostoses, surfing, surfer’s ear, otology, preventive medicine, sports medicine

## Abstract

External auditory exostosis (EAE) has previously only been shown to occur in cold water surfers. We assessed young surfers living and surfing in Queensland, Australia, for EAE in water temp ranges from 20.6 °C (69.1 °F, Winter) to 28.2 °C (82.8 °F, Summer). All participants underwent a bilateral otoscopic examination to assess the presence and severity of EAE. A total of 23 surfers participated with a mean age of 35.4 years (8.3 years) and a mean surfing experience of 20.0 years (9.9 years). Nearly two-thirds of participants (n = 14, 60.9%) had regular otological symptoms, most commonly water trapping (n = 13, 56.5%), pain (n = 8, 34.8%), and hearing loss (n = 6, 26.1%). Only 8.7% (n = 2) of all surfers reported regular use of protective equipment (e.g., earplugs) on a regular basis. The overall prevalence of exostosis was 69.6% (n = 16), and the majority (n = 12, 80.0%) demonstrated bilateral lesions of a mild grade (<33% obstruction of the external auditory canal). This is the first study assessing EAE in young surfers exposed to only warm waters (above 20.6 °C). The prevalence of EAE in this study highlights that EAE is not restricted to cold water conditions, as previously believed. Warm water surfing enthusiasts should be screened on a regular basis by their general medical practitioner and utilize prevention strategies such as earplugs to minimize exposure to EAE development.

## 1. Introduction

It is estimated that there are approximately 35 million surfers worldwide, with 2.7 million in Australia [1]. Given that surfing has been added as a new sport in the 2020 Tokyo Olympics [2] and the development of wave pools, the popularity of this aquatic activity is expected to increase dramatically over the coming years. There have been numerous studies (prospective, retrospective) that have investigated injuries (acute and/or chronic) in recreational and/or competitive surfers, with data attained from online surveys, emergency departments, and medical records [3,4,5,6,7,8,9,10,11,12,13].

Despite the focus of these studies on musculoskeletal injuries and their mechanisms, location, and severity, there have been limited studies on one particular chronic injury, that being external auditory exostosis. Exostosis is colloquially known as surfer’s ear and generally occurs bilaterally [6]. Exostosis is a non-life-threatening medical condition that is benign, irreversible, and believed to be reactive bony outgrowths which develop from the temporal bone from exposure to cold water [14,15,16,17]. The exact mechanism for exostosis is unknown; however, it is widely believed that cold water and cold air stimulate osteoblasts within the temporal bone of the ear and, subsequently, this results in the bony growths providing as a protective mechanism to the tympanic membrane against cold water and cold air [16,18].

Individuals with exostosis may experience water trapping, otitis externa (inflammation of the ear canal), otalgia (earache), and hearing impairment due to the stenosis of the external auditory canal (EAC). Exostosis is classified via otoscopic examination into three grades of severity based upon the percentage of occlusion of the EAC (Figure 1). Grade “1” occludes the EAC by 1% to 33%, Grade “2” occludes the EAC by 34% to 66%, and Grade “3” occludes the EAC by 67% to full occlusion (i.e., 100%) [19]. The only treatment for severe (i.e., Grade 3 and/or highly symptomatic) exostosis remains surgical removal; however, this procedure exposes the surfer to complications which can include tympanic membrane rupture, hearing loss, facial nerve injury and infection [20,21].

Van Gilse [22] was one of the first researchers to report exostosis in cold water swimmers, and shortly thereafter, Fowler and Osmon [23] reported that water temperatures of 17.5 °C resulted in the development of exostosis lesions. The cold water hypothesis was further supported when Kennedy [24] compared coastal populations of surfers and reported that exostosis was more commonly seen in colder waters.

The lifetime prevalence of exostosis in surfers has been reported, ranging from 38% in East Coast US surfers [14] to 80% in Japanese surfers [15] when investigated by otological examination. Hurst [25] reported a lifetime prevalence of 78% in male surfers and 69% in female Victoria (Australia) surfers, with more than fifty percent exhibiting Grade 3 severity. We previously investigated the lifetime prevalence of exostosis in Australian surfers via survey, however, identified only a 3.5% prevalence [26]. At present, there is only one study to date that has investigated the lifetime prevalence of exostosis in warm-water surfers (or swimmers). Kroon and colleagues [14] reported a warm-water prevalence of 15.3% in the East Coast Surfing Championship competitors (Virginia Beach, Virginia); however, the authors defined warm water as a temperature of 16.1 °C (61 °F), and this is below 19 °C, a cut-off temperature that has been previously suggested in the literature [24].

In South-East Queensland (Australia), our local ocean temperature ranges from a minimum of 20.6 °C (69.1 °F) in winter to 28.2 °C (82.8 °F) in summer [27]. Given the popularity of surfing in the local region, we sought to investigate if warm-water only surfers demonstrated exostosis and, if so, determine the severity via otoscopic examination.

## 2. Materials and Methods

We conducted a cross-sectional investigative study where we attempted to recruit 50 surfers aged 18 to 45 years who primarily surfed in the Gold Coast area of Queensland (Australia) (Bond University Human Research Ethics Committee approval number RO15221).

### 2.1. Participants

Participants were required to have surfed year-round with a minimum of five years of current surfing exposure with a minimum of 5 surfing sessions per month. Surfers with ≥ 3 weeks of cold exposure (surfing, swimming, skiing, snowboarding) or a recent history of otological surgery were excluded from participation in this study.

Twenty-three surfers (19 males, 4 females) aged 18 to 45 years (mean age 35.4 SD 8.3 years), who almost exclusively surfed in the Gold Coast area (1 participant surfed in Hawaii (USA) and Bali (Indonesia), both of which have warmer water than the Gold Coast winter ocean temperature), volunteered to participate in this study.

### 2.2. Testing Procedures

Upon arrival at the Water Based Research Unit, participants were provided an information sheet about the study and questions pertaining to the study and its methodologies were answered at this time. Participants then signed informed consent. Next, all participants completed a short questionnaire about their surfing demographics and medical history specific to their ears (i.e., history of exostosis, otitis externa, otalgia, water trapping, and hearing loss). Hearing loss was a subjective symptom when reported by participants; no audiological testing was conducted to quantify this symptom.

Following the completion of the questionnaire, all participants underwent bilateral otoscopic examination with a digital otoscope attached to a laptop computer (Digital MacroView^TM^, halogen HPX fiber-optic otoscope, Welch Allyn^®^, Skaneateles Falls, NY, USA). This device provided live images on the computer, which were later saved digitally in JPEG file format.

Whilst participants were undergoing the otoscopic examination, they were advised if exostosis/exostoses were present and, if so, the severity based upon the one to three scale [19] (Figure 1). All participants received educational information about the disorder and common potential preventative measures (i.e., earplugs, drops).

### 2.3. Statistical Analyses

The normality of all data was assessed by investigating kurtosis, skewness, Q-Q plots, and the Shapiro–Wilk test. Heteroscedasticity was also assessed using Levene’s test for the equality of variances. Where outcome variables were categorical, results were summarized as frequencies and percentages (absolute). Data are presented as mean (SD) or percentage throughout unless noted. A bivariate Spearman rank-order correlation was utilized to assess the associations between age, years surfing, and winter surf exposure and the severity of exostosis. Statistical analyses were competed using IBM SPSS Statistics for Windows statistical software (Version 25.0, IBM Corp., Armonk, NY, USA).

The required minimum sample size for adequate study power was computed. This analysis was conducted for a study design comparing the study group to a population incidence based on a dichotomous/binomial endpoint (exostosis or no exostosis). With the variables utilized to calculate minimum sample size set at alpha = 0.05, power = 80% (giving beta of 20%), and an anticipated population incidence of 38% (the lowest reported incidence in the literature in otoscopic studies), the minimum sample size for adequate study power was calculated at 10 subjects (5).

## 3. Results

There were no significant differences between genders; males were slightly older than females (8.1%) and reported more years surfing (35.6%). Six of the participants also swam, two participated in stand-up paddle boarding, and two participants participated kite surfing in the Gold Coast area. Participants had been surfing for an average of 20.0 years (range 6 to 42) and had skill levels (according to the Hutt scale [28]) from beginner (4.3%) to top amateur (21.7%, able to consecutively execute advanced maneuvers). The majority (95.7%) were riding short boards. The participant demographics are listed in Table 1.

Regarding the diagnosis of exostosis, participants were identified with an overall lifetime prevalence of 69.6% of exostosis (three females and 13 males). When we evaluated the lifetime prevalence of exostosis, the youngest participant identified was a female of 27 years of age. The youngest quartile (≤31 years) relative lifetime prevalence was 33.3%, the 2nd quartile (aged 32 to ≤37 years) relative lifetime prevalence was 83.3%, the 3rd quartile (aged 38 to < 41 years) was 80.0%, and the oldest quartile (≥42 years) had a relative lifetime prevalence of 83.3% (Figure 2). Of those diagnosed with exostosis, the majority (80.0%) demonstrated bilateral exostosis (i.e., exostosis was identified in both the right and left ear) with a severity of Grade 1 (30.4%) (Table 2).

There was a significant correlation between age and the severity of exostosis for both ears (right r^2^ = 0.428, *p* = 0.042; left r^2^ = 0.606, *p* = 0.002) and years surfing and severity of exostosis for both ears (right r^2^ = 0.538, *p* = 0.08; left r^2^ = 0.613, *p* = 0.003). Additionally, there was a significant (*p* = 0.003) positive correlation between the severity of exostosis in the right ear and the severity of exostosis in the left ear (r^2^ = 0.591, *p* = 0.001), There was, however, no significant correlation between winter surf exposure and the right and left ear exostosis severity (r^2^ = 0.231, *p* = 0.588; r^2^ = 0.176, *p* = 0.987, respectively).

Approximately two thirds of participants (60.9%; male = 11, female = 3) reported experiencing regular otological symptoms, the most common of which was water trapping (56%; male = 11, female = 2), pain (34.8%; male = 6, female = 2), and hearing loss (26.1%; male = 4, female = 2). With regard to exostosis prevention, the majority (69.6%) were aware of prevention methods, however, only approximately one-third (34.8%; males = 6, females = 2) actually reported utilizing any form of prevention for exostosis. The most commonly reported prevention method was earplugs (26.1%) followed by Blu-tack^®^ and a combination of earplugs and ear drops (4.3% each).

## 4. Discussion

The primary purpose of this study was to determine if exostosis, which was previously believed to be limited to cold-water only surfers, existed in warm water Australian surfers and, if so, determine the severity via otoscopic examination. Our main finding indicated that exostosis is indeed present in warm-water surfers, and its lifetime prevalence in our participants was similar to that previously reported in cold water surfers [15].

### 4.1. Lifetime Prevalence of Exostosis

We identified a lifetime prevalence of exostosis of approximately 70% in our cohort of participants, which were warm-water surfers in South-East Queensland (Australia, winter temperature ranged from 20.6 °C (69.1 °F) to 28.2 °C (82.8 °F) in summer). These findings are similar to those reported in cold-water surfers by Umeda et al. [15], who identified a prevalence of 80% exostosis in Japanese surfers and findings similar to Hurst and colleagues [25], who reported a prevalence of exostosis of 75% in male surfers and 69% in females surfers in Victoria, Australia (13.6 °C in Winter to 20.3 °C in summer) [27]. Our findings are also similar to Attlmayr and Smith [16], who reported a prevalence of 63.8% in Cornish surfers (water temperature 8.4 °C in winter to 19.3 °C in summer) [27]. Chaplin and Stewart [29] investigated the prevalence of exostoses in New Zealand surfers (n = 54) and surf life savers (n = 38) in Dunedin (water temperature 10.0 °C in winter to 14.0 °C in summer) and reported an overall prevalence of 73.0%. Additionally, all surfers with 10 years or more surfing experience had some evidence of exostosis.

It is interesting to note that when we evaluated exostosis via survey, as opposed to otoscopic examination, we found a lifetime prevalence of only 3.5% in Australian surfers [26] and more recently [30] 28.9% in New Zealand surfers. Chaplin and Stewart [29] reported a lifetime prevalence of exostosis of 73.0% in New Zealand surfers, with one half presenting with a Grade 2 or Grade 3 severity. We therefore believe that otoscopic examination is required to determine the presence (and severity) of exostosis.

### 4.2. Exostosis in Other Aquatic Activities

Although exostosis appears to be well investigated in surfers, exostosis has also been reported in other aquatic activities, such as diving [31,32,33,34], swimming [35,36,37], and kayaking [38,39]. Exostosis has been reported in divers to have a prevalence ranging from 26% in US navy divers [32] to 85.7% in Japanese Matsu divers [31]. Swimmers and kayakers (69.5% to 79%) appear to have a similar prevalence of exostosis.

With regard to investigating the prevalence of exostosis in warm water, Kroon et al. [14] is the only study that reported the prevalence of exostosis in warm-water surfers (lifetime prevalence 31%); however, they used a warm-water temperature of 16.1 °C (61 °F). Our study’s ocean temperatures were 28.0% warmer in winter (20.6 °C, 69.1 °F) and considerably warmer in summer (28.2 °C, 82.8 °F). We believe that this is a better reflection of warm water, based on an anthropological study where the temperature of 19 °C (66.2 °F) was suggested as a cut-off point [24]. However, Deleyiannis et al., [17] did postulate that Californian surfers in warmer waters may simply require more time (i.e., exposure) for exostosis to occur.

This study is the first to identify exostosis in warm water, and as such, the mechanism for exostosis is unknown. Previously, it was believed that cold water and cold air stimulated osteoblasts within the temporal bone of the ear and, as a result, bony growths developed as a protective mechanism. However, based on our high prevalence of exostosis in warm-water surfers, the threshold of water and air temperature to stimulate the bony outgrowths appears to be warmer than previously believed. Deleyiannis and colleagues [17] postulated that surfers in warmer water may just require a longer period of exposure to develop exostosis. The critical temperature for the development of exostosis, however, is yet to be determined.

With regard to gender and lifetime prevalence of exostosis, 68.4% of all male participants and 100% of all female participants demonstrated exostosis. These values are similar to that previously reported by Hurst and colleagues [25] in cold-water surfers in Victoria (Australia). Participants from our youngest (18 years) to our oldest (45 years) demonstrated exostosis.

We reported a significant correlation between years surfing and the severity of exostosis; this is in agreement with Attlmayr and Smith [16], who investigated the prevalence and severity of exostosis in Cornish surfers and also identified a significant positive correlation. Attlmayr and Smith [16], however, also reported a significant correlation between the severity of exostosis and cold-water exposure, whereas we found no relationship between Winter surf exposure and the severity of exostosis.

Despite the high prevalence of exostosis in surfers, we could not identify any study that evaluated the effectiveness of protection strategies for exostosis. Timofeev and colleagues [40], in their surgical follow-up of patients with exostosis, reported that wearing earplugs prolonged the recurrence-free time period by a factor of five compared with no ear protection. Chaplin and Stewart [29] reported no difference in the severity of exostosis in cold water surfers that wore protective earplugs versus those who did not. They also commented that there was no difference in the severity of exostosis in those surfers who used either a chemical drying agent in their ears and those that did not. However, further work is required to determine the long-term efficacy of prevention strategies for exostosis.

With regard to otological symptoms, a number of studies did not report otological symptoms [14,17,41] or specifics [29,42]. Nakanishi et al. [43] found that the most common otological symptom was water trapping, and this occurred more frequently in surfers with grade 2 or more severe exostosis. They did report that there was no correlation between the severity of exostosis and other symptoms, and hearing loss was not an accompanying symptom even in grade 3 exostosis. Attlmayr and Smith [16] found that approximately one-third of their subjects reported recurrent ear infections or otalgia and the presence of symptoms increased significantly with the severity of exostosis.

Our principal strengths to this study were our methodology and inclusion criteria. Regarding methodology, we utilized a digital otoscopic attached to a notebook portable computer. The otoscope is the gold standard [19] for identifying if exostosis was present and, if so, to determine its severity. Additionally, the otoscope was connected to a laptop computer with high-resolution screen, which assisted the investigator, an experienced Sports Medicine and emergency medicine physician, with detailed analysis of identifying the presence of exostosis and quantifying its severity. Our inclusion criteria for defining warm-water surfers was quite stringent and allowed us to reduce possible confounding variables on the development of exostosis.

The primary limitation to this study was its small sample size (n = 23). Although we initially (a priori) computed the sample size numbers required, the small sample size limits the generalizability of our findings. Our stringent inclusion criteria also limited our ability to attain a larger sample size. Additionally, the limited geographic area in which we attained our sample makes generalizability limited. Although we identified exostosis in warm-water surfers, warmer water (i.e., further north in Australia, Indonesia, Fiji, Caribbean, etc.) may or may not be a sufficient stimulus to result in exostosis. This is yet to be investigated and represents an area for future research.

## 5. Conclusions

This study is the first to identify the lifetime prevalence of external auditory exostosis in exclusively warm-water surfers and found the prevalence to be similar to cold-water surfers. These findings have a significant health impact, as there are significantly more surfers now vulnerable to the development of exostosis. Warm-water surfing enthusiasts should, therefore, be screened on a regular basis by their general medical practitioner and utilize prevention strategies, such as earplugs, to minimize exposure to external auditory exostosis (EAE) development. Given the increased susceptibility to EAC, there is a need for research into the efficacy of proposed preventative strategies.

## Figures and Tables

**Figure 1 sports-08-00017-f001:**
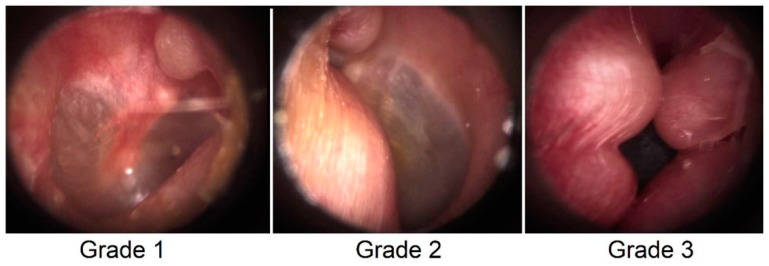
Otoscopic images identifying the grades of exostosis.

**Figure 2 sports-08-00017-f002:**
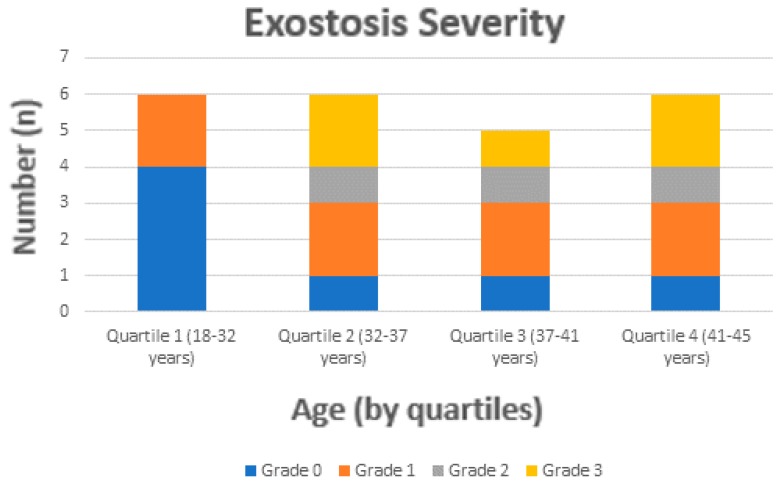
Exostosis severity by age quartile.

**Table 1 sports-08-00017-t001:** Participant’s demographics (vales are mean SD or percentage).

Parameter	Group (n = 23)	Males (n = 19)	Females (n = 4)	*p* Value
Age (years)	35.4 (8.3)	35.9 (7.5)	33.0 (12.7)	0.06
Surfing experience (years)	20.4 (9.9)	21.4 (8.23)	13.8 (10.9)	0.17
Winter surfing sessions (number)	14.0 (8.3)	12.9 (8.3)	19.0 (6.8)	0.19
Winter surfing sessions (hours/session)	1.7 (0.6)	1.8 (0.7)	1.6 (0.3)	0.11
Surfing stance:				
• Regular (left leg forward, %)	87.0	84.2	25.0	–
• Goofy (right leg forward, %)	13.0	15.8	75.0	–
Board type:				–
• Short board (%)	95.7	100.0	75.0	
• Long board (%)	4.3	0.0	25.0	

**Table 2 sports-08-00017-t002:** Participant’s demographics with regard to exostosis (values are mean SD or percentage).

Parameter	Group (n = 23)	Males (n = 19)	Females (n = 4)
Exostosis identified (%)	69.6	68.4	75.0
Exostosis by Age Quartile (%)			
• ≤ 31 years	66.70	75.0	50.0
• 32 to ≤ 37 years	83.3	83.3	0.0
• 38 to ≤ 41 years	80.0	80.0	0.0
• ≥ 42 years	83.3	83.3	100.0
Exostosis Severity (1 to 3)			
• no visable exostosis (%)	41.3	42.1	16.7
• Grade 1 (%)	30.4	26.3	66.7
• Grade 2 (%)	10.9	10.5	16.7
• Grade 3 (%)	17.4	21.1	0.0
Exostosis Bilateral			
• Yes (%)	80.0	76.9	100.0
Exostosis location and severity			
• Right ear	52.2	52.6	50.0
Grade 1 (%)	41.7	30.0	100.0
Grade 2 (%)	25.0	30.0	0.0
Grade 3 (%)	33.3	40.0	0.0
• Left ear	65.2	63.2	75.0
Grade 1 (%)	60.0	58.3	66.7
Grade 2 (%)	13.3	8.3	33.3
Grade 3 (%)	26.7	33.3	0.0

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
