# Peer review of "The Prevalence and Severity of External Auditory Exostosis in Young to Quadragenarian-Aged Warm-Water Surfers: A Preliminary Study"

_sports, 2020, doi:10.3390/sports8020017_

Round 1
Reviewer 1 Report
The authors of "The prevalence and severity of EAE in young warm water surfers: a preliminary study" present a well-designed and useful study. The main message of the manuscript is that surfing in warm waters does not protect from EAE. Also, the authrs suggest that warm water surferfs should undergo regular otological screening and should use ear protection while surfing. I see only a couple of minor issues that should be addressed by the authors. The first issue regards the sample size, which, despite being properly calculated, is relatively small. Therefore, in the abstract, I suggest providing the numbers followed by the percentages instead of using percentages only. The second issue regards the phrase "young swimmers" in the title. Young adults are considered to be between 18 and 25 years of age (https://www.reference.com/family/age-young-adult-c5b1a8a6553b26d9). Other sources refer to young adults as between 21 and 29 years of age (https://www.ncbi.nlm.nih.gov/books/NBK13167/). I think that calling a 45-years old young person is not appropriate and I suggest changing the title respectively.
Reviewer 2 Report
I would like to thank you for submitting and give me the opportunity to review this interesting work about the prevalence and severity of exostosis in warm water surfers. I hope my comments will help to improve the quality of the manuscript in some way.
In general, the manuscript is well structured and clear. Introduction and methods sections are well structured and give a comprehensive context of the manuscript. However results section need some improvements and discussion section is weak, authors should go deeper into the interpretation of the main results obtained.
Below I will make some specific comments on each section that should be carefully reviewed.
My first concern is about the title of the paper, it appears the word “young” but your n mean age is 35.4 (8.3) years. Please consider removing the word “young” from the title.
Abstract
In line 24-25 it appears a 57.8% that it do not appear later in the manuscript, please check if it is correct. It is also a bit confusing to read that the prevalence of exostosis was 69.6% but then you say that only 57.8% demonstrated some degree of exostosis.
Methods
In statistical analysis authors carry out a Kolmogorov-Smirnov normality test with n=23, it may be more appropriate to use Shapiro-Wilk test (n<50).
Results
This is a personal consideration, but from my view point, the first section (from line 127 to line 139) of the results should be in participants (methods) section, because it helps the reader to better understand the procedures.
Table 2. When authors show exostosis location and severity, the percentages shown are a bit confusing. For example, in the complete group in Right ear they found 52.2 while in left ear they found 65.2, this add up to 117.4. The same occur when we focus on grade 1 (right ear) in the whole group it appears 100% while when we add males and women percentages they add up 65.8%. Please clarify the whole exostosis location and severity section of this table.
The results that appear from line 150 to line 154 should appear in dispersion graphs to lead to a better understanding of them.
As well as the results that appear from line 158 to line 163, that should appear in a table or a figure divided in group, male and female, as table 2.
Discussion
In this section, there is a lack of results interpretation. Authors only compare their results with previous studies, but an explanation of those results found is needed. In this sense, I would like to understand why exostosis appear in warm water, while it is believed that this injury have a greater prevalence in cold water.
Correlations between experience and severity of exostosis and winter surf exposure and right/left ear exostosis severity are not discussed.
Otological symptoms are not discussed.
Conclusions
The conclusions are very general and they seem more an interpretation of the results than an actual conclusion of the work. From my viewpoint, the conclusions of the abstract are quite better than those in conclusions section.
Round 2
Reviewer 2 Report
Authors have adequately reviewed and corrected all the comments indicated in the previous version.
Author Response
We thank Reviewer 2 for his/her constructive comments to improve our manuscript.